# Data Analytics of Broiler Growth Dynamics and Feed Conversion Ratio of Broilers Raised to 35 d under Commercial Tropical Conditions

**DOI:** 10.3390/ani13152447

**Published:** 2023-07-28

**Authors:** Gustavo A. Quintana-Ospina, Maria C. Alfaro-Wisaquillo, Edgar O. Oviedo-Rondon, Juan R. Ruiz-Ramirez, Luis C. Bernal-Arango, Gustavo D. Martinez-Bernal

**Affiliations:** 1Prestage Department of Poultry Science, North Carolina State University, Raleigh, NC 27695-7608, USA; gustavoquintana22@gmail.com (G.A.Q.-O.); camilaalfaro2164@gmail.com (M.C.A.-W.); 2Grupo BIOS Inc., Envigado 055420, Antioquia, Colombia; juan.ruiz@grupobios.co (J.R.R.-R.); luis.bernal@grupobios.co (L.C.B.-A.); gustavo.martinez@opav.co (G.D.M.-B.)

**Keywords:** data analytics, growth models, non-linear equation, live performance, broilers

## Abstract

**Simple Summary:**

Although weekly performance parameters in broiler systems are usually collected for monitoring, it is not common to use data to determine the impact of chicken growth parameters on feed conversion ratio or to predict body weight at slaughter age. Additionally, husbandry variables or environmental conditions might lead to developing strategies within poultry companies to overcome different in-field challenges. Understanding the growth dynamics of chickens subject to determined conditions could help nutritionists and managers improve the decision-making processes to achieve desired performance results. The objective of this study was to estimate and compare the growth pattern of males and females Ross 308 AP broilers reared under commercial tropical conditions with controlled feeding to determine conditions that optimize feed conversion ratio. The results indicated that the most efficient broilers were those subject to a more significant feed control between the second and the fourth week of age, with an increase in the feed allowance during the last week compared to those birds subject to a constant controlled feeding throughout the production cycle. It suggests that after chickens were conditioned to feeding control, a greater feed allowance during the week before the slaughter could result in heavier chickens at 35 d and still be more efficient.

**Abstract:**

Data collection is standard in commercial broiler production; however, growth modeling is still a challenge since this data often lacks an inflection point. This study evaluated body weight (BW) dynamics, feed intake, BW gain, feed conversion ratio (FCR), and mortality of broiler flocks reared under commercial tropical conditions with controlled feeding to optimize FCR. The data analyzed included performance records of 1347 male and 1353 female Ross 308 AP broiler flocks with a total of 95.4 million chickens housed from 2018 to 2020. Decision trees determined high- and low-feed-efficiency groups using FCR at 35 d. Logistic, Gompertz–Laird, and von Bertalanffy growth models were fitted with weekly BW data for each flock within performance groups. The logistic model indicated more accurate estimates with biological meaning. The high-efficiency males and females (*p* < 0.001) were offered less feed than the low-efficiency group and were consistently more efficient. In conclusion, greater feeding control between the second and the fourth week of age, followed by higher feed allowance during the last week, was associated with better feed efficiency at 35 d in males and females. Additionally, models demonstrated that a reduced growth rate resulted in heavier chickens at 35 d with better feed efficiency and greater BW gain.

## 1. Introduction

Broiler feed conversion ratio (FCR) is affected by growth dynamics [1,2], which is expressed as a relationship between body weight (BW) or feed intake (FI) with age. Growth curves are non-linear models traditionally used to depict the relationship between growth parameters and age [3,4,5,6,7]. These models may include the Gompertz function [8], the Gompertz–Laird form [9], and some other Gompertz variations [10], as well as the Logistic [11] and von Bertalanffy functions [12], among others [3,6,13].

Under commercial settings, broilers are typically raised with ad libitum feed allowance. However, controlled feeding programs have been employed for several years in regions with high altitudes to reduce broilers’ growth rate and decrease high mortality rates derived from metabolic issues such as ascites and sudden death syndrome [14,15,16,17]. On the other hand, Zuidhof et al. [18] indicated that BW gain had increased by about 400% while the FCR has been reduced by 50% in the last 50 years, thus changing the growth dynamics of broilers. The constant genetic selection for the improvement of broilers’ feed efficiency added up to the strategies employed in commercial operations to raise broilers has led many authors to compare the chickens’ growth curves from genetically selected fast-growing [5,19] and native [20,21,22] broiler strains, in sexed and unsexed systems [4,23,24]. Nevertheless, reports about the growth of broiler chickens subjected to feeding control and raised under commercial tropical conditions are still scarce.

Conflicting results have described how different feeding programs may affect the FCR and BW at slaughter age [1,15,25,26]. Some authors have suggested that using controlled feeding programs with up to 20% reduction of the ad libitum intake during the second week of age [1,15] or the entire cycle [26] resulted in better FCR early in life with no significant differences in BW at slaughter age (36 d) due to compensatory growth [27,28]. Contrarily, Dissanayake and David [25] showed that chickens with feeding controlled 10, 20, and 30% from the ad libitum intake did reduce both the FCR and the BW, except for the treatment with 10% restriction that increased the BW at 42 d. In this scenario, growth curves play an essential role by predicting live performance responses based on the growth dynamics and providing estimates with biological meaning that allow comparing flocks of different sexes and feed efficiency as described by previous research reports under controlled conditions [25,26,27,28]. These properties could be beneficial to be used under commercial conditions to design strategies to predict performance or evaluate the effects of nutritional and management factors [3,6,29].

On the other hand, the lack of predefined groups or treatments as independent variables for unstructured data analysis is limiting and concerning under commercial operations [30,31]. Therefore, data mining tools are needed to unveil patterns when large amounts of data are presented. Within the available tools, the classification and regression tree algorithms, also known as decision trees, allow the creation of subgroups based on categorical or continuous predictors [32,33]. Partition trees analyze the independent variable to determine the ideal split point of the dataset by examining the relationship between the variables with Pearson’s Chi-square. However, because this statistical test relies on the sample size, *p*-values tend to be extremely low or significant and must be corrected by the inverse of the *p*-value or LogWorth [34]. It indicates that the subgroups produced in a decision tree differ statistically. This partition technique could determine groups of low (LE) and high (HE) efficiency of feed utilization in a dataset of a commercial broiler integration to determine the root causes of performance. The objective of this study was to estimate and compare the growth pattern of males and females Ross 308 AP broilers according to two categories of FCR at 35 d, HE, and LE and reared under commercial tropical conditions with controlled feeding to determine conditions that optimize FCR.

## 2. Materials and Methods

### 2.1. Data and Database Description

One database was obtained from a broiler integration located in Colombia. The database contained BW, FI, FCR, and mortality records of 1649 male and 1406 female Ross 308 AP broiler flocks subject to controlled feeding from the Aviagen [35] recommendations. However, data were cleaned, and flock records with missing data were removed. Thus, a total of 1349 male and 1353 female flocks presented complete data collected weekly from flock placement until 35 d. The database was organized so that each column represented a performance variable and each row a broiler flock. Weekly BW corresponds to averages of samples of 1% of the population per broiler house, as it is usually evaluated in industrial settings. Feed was weighed and offered according to the feed allocation plan designed for each farm. The cumulative percentage of mortality for each week was calculated. The data analyzed represented approximately 95 million chickens raised between 2018 and 2020 in 86 farms under commercial tropical conditions with controlled feeding across Colombia’s three major poultry production regions. In this operation, eggs were incubated using multistage machines in two company-owned hatcheries. At hatch, chicks were sexed, boxed, and transported to farms located up to 531 km away from the hatcheries and at a mean altitude of 1381 ± 449 m above sea level. Subsequently, hatchlings were placed in litter-based open-sided broiler houses with rice hulls or wood shavings and fed pre-starter (150 g/bird), starter (900 g/bird), and grower (1643 ± 145 g/bird) diets up to 35 d. Stocking density was 12.74 ± 1.12 chickens or 28.5 kg/m^2^. The database was organized in Excel^®^ (Microsoft Corporation, version 2019, 16.0, Redmond, WA, USA) and saved as a comma-separated value file to be imported into the statistical software. The data-cleaning process included comparisons with physical records and the recalculation of parameters to determine the accuracy of all variables.

### 2.2. Statistical Analysis

Data were processed following the flow presented in Figure 1. Data were analyzed using R (R Core Team, 2021) in RStudio (RStudio Team, Boston, MA, USA) and JMP Pro 15 (SAS Institute, Cary, NC, USA).

#### 2.2.1. Data Partition

Decision trees [32,33] were fitted to determine two efficiency groups based on sex and FCR at 35 d. Partition trees included flock ID as a predictor and FCR at 35 d as the response.

#### 2.2.2. Additional Parameters of Live Performance

Percentage data for the week and cumulative mortality were transformed into the arcsine square-root percentage for analysis. Weekly BW, FI, FCR, week mortality, and cumulative mortality were also subject to a one-way ANOVA with the efficiency group as the main effect and mean separation with a Student’s *t*-test.

#### 2.2.3. Growth Models

Three growth models, Logistic [11], Gompertz–Laird [9], and von Bertalanffy [12], were fitted both to each flock and all data within the efficiency groups with weekly data to estimate the relationship between age and BW. The following equations describe the models:**Logistic:**(1)$$y=W_{A}/[1+\exp\;-K(t-ti)]$$
where,*W_A_* = Asymptotic response;*K* = Exponential growth rate;*ti* = Age at the inflection point.**Gompertz–Laird:**(2)$$y=W_{0}*\exp\;[(L/K)(1-\exp\;-Kt)]$$
where, *W*_0_ = Initial response;*K* = Rate of exponential decay;*L* = Initial growth rate.**von Bertalanffy:**(3)$$y=W_{A}*(1-B*exp{\left(-K*t)\right)}^{3}$$
where,
*W_A_* = Asymptotic response;*K* = Maximum relative growth;*B* = Integration constant.

The response in which the maximum growth rate is observed, also known as *Wi* and maximum increment (*MI*), were estimated using the parameters of the logistic model as reported by Al-Nasrawi [4] with the equations listed below:(4)$$Wi=W_{A}/2$$
(5)MI=K∗Wi/2

Likewise, the *Wi* and age at the inflection point (*ti*) from the von Bertalanffy model were calculated [36] as follows:(6)ti=ln(3B)/K
(7)$$W_{i}=W_{A}*8/27$$

From the Gompertz–Laird model estimates, *ti* and asymptotic response (*W_A_*) were determined [3] with the following equations:(8)ti=(1/K)∗log (L/K)
(9)$$W_{A}=W_{0}\;exp\;(L/K)$$

All growth models were fitted using the R stats package’s non-linear least squares (*nls*) function. Coefficient of determination (R^2^), Akaike’s information criterion (AIC), Bayesian information criterion (BIC), number of iterations, and root mean square error (RMSE) were calculated using the broom R package [37] and utilized as goodness-of-fit metrics to evaluate model performance as described in previous reports [3,4,5,6]. The best model criteria included the highest R^2^ and the lowest AIC, BIC, RMSE, and the lowest iterations. Significance was determined at an alpha level of 0.01.

Parameter estimates from the model of each flock were analyzed as dependent variables using a one-way ANOVA with efficiency as an independent variable and mean separation with Student’s *t*-test. Correlation and regression analyses were conducted to determine relationships among parameter estimates [4,6,13,23,29,31].

## 3. Results

### 3.1. Data Partition

Results of the decision tree for males are presented in Figure 2 and for females in Figure 3. The decision tree for males included seven splits and classified (R^2^ = 0.93, RASE = 0.157) the flocks into two feed efficiency groups (HE and LE). The HE group contained 51 flocks with an average FCR of 1.371 at 35 d, while the LE group had 105 flocks with an average FCR of 1.527. For females, the decision tree contained eight splits and determined (R^2^ = 0.95, RASE = 0.011) one HE group comprising 31 flocks with an average FCR of 1.426 and the LE group with 128 flocks and FCR of 1.526 at 35 d.

### 3.2. Live Performance

A summary of live performance results for male and female flocks is presented in Table 1. Male BW differed (*p* < 0.001) between efficiency groups at all ages except for 7 d. Males from the HE group were heavier (*p* < 0.001) at hatch and 35 d but lighter (*p* < 0.001) at 14, 21, and 28 d. In contrast, efficiency groups in the females were only significantly different (*p* < 0.05) at 14 and 35 d. The LE females had higher BW at 14 but lower at 35 d. Although FI was controlled in the integration with a pre-planned table, there was variation in the amount of feed offered between farms and flocks. However, differences (*p* < 0.001) in FI were observed during the whole production cycle. On average, the HE males and females (*p* < 0.001) were offered 8.52% and 4.64% less feed than the LE group. Similarly, LE chickens from both sexes were consistently (*p* < 0.001) less efficient than their counterpart from 7 to 35 d. Significant differences (*p* = 0.001) were observed in week mortality only in males from 8 to 14 d. Chickens from the LE group presented (*p* = 0.001) mortality 0.22% higher than HE chickens (0.81 vs. 0.59%). Consequently, male cumulative mortality up to the second (*p* < 0.001) and the third (*p* = 0.001) week of age were significantly different between efficiency groups. LE males resulted in cumulative mortalities of 0.31 and 0.38% higher than the HE flocks at 14 and 21 d. No other differences were detected in males from the third week onwards or between female efficiency groups on the week and cumulative mortality.

### 3.3. Growth Models

Parameter estimates for all models are presented in Table 2. Significant differences (*p* < 0.001) were detected between parameter estimates from male efficiency groups in the Logistic, Gompertz–Laird, and von Bertalanffy models. All parameter estimates from the Logistic model were higher (*p* < 0.001) in HE male chickens except for *K* (*p* < 0.001), which was 0.008 lower (0.122 vs. 0.114). It indicated that the LE males reached the *Wi* 4 days earlier (*p* < 0.001) and were 361 g lighter (*p* < 0.001) than the HE group. Additionally, males from this group presented an average MI of 14 g/d higher than their counterparts (103 vs. 89 g/d). The Gompertz–Laird models predicted heavier HE chickens (*p* < 0.001) at hatch and asymptotic age (*p* < 0.001) with a longer *ti* (*p* < 0.001). Significant differences in *L* and *K* indicated LE males presented faster initial growth rates and rates of exponential decay (*p* < 0.001). Similarly, the von Bertalanffy model for males indicated higher (*p* < 0.001) parameters for HE males compared to LE chickens, except for *K*, which was greater (*p* < 0.001) in the LE group. Although significant differences (*p* < 0.001) were detected in *Wi*, the results indicated greatly overestimated values for *Wi* in both male and female von Bertalanffy models.

On female BW, estimates from the logistic model resulted in HE chickens being 363 g heavier at asymptotic age (*p* < 0.001) compared to the LE group. HE female chickens reached the *ti* at 32 d, while LE flocks exhibited the maximum daily growth at 30 d. However, the difference in the *Wi* between HE and LE was estimated (*p* < 0.001) in 181 g (1510 vs. 1329 g) with 9 g/d more (*p* < 0.001) *MI*, respectively. No significant differences (*p* > 0.05) were observed in *K*. Similarly, Gompertz–Laird model estimates showed no significant differences (*p* > 0.05) between efficiency groups in *W*_0_ and *L* means. Significant effects of efficiency were detected in *K*, *ti*, and *W_A_* from the Gompertz–Laird model. LE females presented a higher rate of exponential decay (*p* < 0.001) but lower (*p* < 0.001) *ti* and *W_A_*, which indicates that females reached the inflection point younger and resulted in a lighter weight at slaughter age, respectively (Table 2). Significant efficiency effects were also observed in all parameters from the von Bertalanffy models. All estimates from LE chickens were lower (*p* < 0.001) except for *K*, which showed a higher estimate (*p* < 0.001) compared to the LE group. Strong (*p* < 0.001) negative correlations (r = −0.86) were detected between model parameters associated with growth rates and age at *Wi* regardless of model, sex, or efficiency group.

Predicted BW and BW gain from all three models (Figure 4) indicated that LE males presented a daily BW gain higher than HE chicken flocks up to 24 d. However, HE male chicken flocks exhibited a lower decay rate, resulting in superior daily BW gain after 24 d and heavier chickens from 32 d onwards. In contrast, LE females were slightly over their counterparts up to 18 d, but as observed in the males, the difference in growth rate was enough to produce heavier female chickens from 25 d.

Goodness-of-fit metrics (Table 3) from models that fit all data by efficiency and sex resulted in similar R^2^, AICc, and BIC values. However, the von Bertalanffy model presented the greatest number of iterations for both efficiency groups and sexes compared to the Logistic and Gompertz–Laird models. On average, the RMSE from the Logistic models was 3.39 higher than the other two models. In contrast, residual analyses for HE male chickens’ data (Figure 5) indicated that all three models overfit the response at 0 and 28 d. The von Bertalanffy model presented the least error compared to the logistic and Gompertz–Laird models at both ages. At 7 and 21 d, all models underfit the BW, except for the logistic model, which was 2.94 g above the mean BW compared to the Gompertz–Laird and von Bertalanffy models, which were 7.7 and 9.0 g below for that week, respectively. On females, all models overestimated the bodyweight at hatch at 28 and 35 d in the HE group. From these results, the von Bertalanffy model presented the lowest residuals at hatch and 28 d but the highest at 35 d.

Similarly, at 14 and 21 d, all models underfit the response. The von Bertalanffy model was only −1 and −5 g, far from the response, while the Gompertz–Laird was −8 g at both ages and the logistic −19 and −22 g at both ages 14 and 21 g, respectively. In contrast, all models for the LE female group underfit the response. The von Bertalanffy model resulted in the lowest error at hatch and 7 d, while the logistic model was more accurate from 14 to 28 d but presented the highest error at 35 d.

## 4. Discussion

### 4.1. Live Performance

In the present study, HE male chicken flocks had a BW at hatch 2 g lower than LE chickens, resulting in heavier broilers at 35 d and more efficiency throughout the production cycle. Several authors have described the BW at hatch as a trait highly related to egg weight [38,39,40], breeder age [41,42,43], and incubation conditions [44,45,46]. Thus, significant differences in BW might be observed at hatch. However, Duman and Şekeroğlu [40] reported that differences in BW at hatch disappeared after the first week of age and affected neither BW nor FCR at 35 or 39 d in Ross 308 broilers. In contrast, Nazligül et al. [47] indicated that birds hatched from smaller or bigger eggs could develop normally due to compensatory growth with intensive care. Duan-yai et al. [23] evaluated three available commercial strains of broilers in Australia (C, R, and I). They determined that the male chickens from the I strain, which were the lightest at hatch, were 141 g heavier at 35 d than the other two strains (1922 vs. 1781 g) but 379 g less at 105 d (5239 vs. 5618 g).

Some poultry-producing companies utilize quantitative controlled feeding in broilers to mitigate the adverse effects of lower oxygen availability and cool environmental temperature as elevation increases in the mountains and consequently avoid high mortality due to ascites [15,48]. Additionally, this approach has been employed as a strategy to reduce other metabolic disorders and bone developmental issues related to the rapid growth of modern chickens [16,17]. Although all animals in the present study were feed-controlled throughout the cycle compared to the Ross 308 AP recommendations [35], the level of feed control observed was greater in both males and females from the HE groups than in the LE chickens.

Some studies have described the early feeding restriction [1,15,17] as a method that limits the amount of feed allowance only during the second week of life. Mohammadalipour et al. [15] indicated that Ross 308 male chickens were significantly more efficient during the second and third week of age when subjected to a 40% feed restriction from 7 to 14 d. Birds were under normal environmental conditions (33 °C, 30 °C, and 27 °C for weeks 1, 2, and 3, respectively). Afterward, no significant effects of feed control were observed in that study. Similarly, Buntzen et al. [1] demonstrated that Cobb 500 male and female broilers fed at 80% of the ad libitum intake from 8 to 16 d were significantly lighter from 16 to 21 d. Still, in experimental conditions, chickens could exhibit compensatory growth reaching BW similar to the control treatment from 28 d onwards. In that study, no significant differences among treatments were reported either in BW for both sexes at 42 d or in FCR from 1 to 42 d between these two treatments. Khurshid et al. [26] showed that reducing the amount of feed offered between 10 and 20% of the ad libitum intake from 1 to 36 d resulted in lighter broilers at 36 d compared to those chickens without feeding control. Still, there was no difference in FCR among controlled feeding treatments (5, 10, 15, and 20%).

In contrast, Dissanayake and David [25] reported that using a feed allowance reduced 10, 20, and 30% from the ad libitum group; the FCR was improved when the feed restriction was increased in all periods evaluated (0 to 21, 22 to 42, and 0 to 42). However, the best performance was observed at a 10% restriction that lowered the FCR but increased the BW. Bordin et al. [2] also reported that chickens subject to feed restriction of 20% from 10 d presented the lowest FCR. Additionally, they observed that the daily BW gain and flock uniformity did not vary between chickens restricted at 10% or fed ad libitum. According to these studies, chickens could preserve a greater feed efficiency for one week more after the established controlled feeding. Still, more extended periods of feed restriction could condition chickens to be more efficient. It has also been described that feed restriction in broilers could improve the ileal digestibility of dry matter, crude protein, and energy [49], as well as produce morphological changes in epithelial cells from the small and large intestines that could modify and enhance the transport of nutrients [2,50,51].

This study showed that while LE males had a consistent controlled feeding with an average of 17.25% lower intake compared to the Ross 308 AP recommendations from the second week onwards, the HE males were subject to a feed control of 25.45% on average from the second week of age up to four weeks and then a reduction in the restriction of 1.62% in the last week (25.45 vs. 23.83%). It seems that after chickens were conditioned to feeding control, a greater feed allowance during the previous week to slaughter could result in heavier chickens at 35 d and still be more efficient. Similarly, LE female flocks resulted in a feed control of 18.1% from the second week onwards. HE females demonstrated a feed control of 22.5% between two and four weeks of age and were later offered 3.38% more feed during the last week.

### 4.2. Growth Models

Significant differences between efficiency groups were observed in the present study’s male and female model estimates. Parameters from all models were higher in HE groups than in LE groups, except for estimates associated with lower growth rates (*K*) for all three models. However, *W_A_* widely differed among models, resulting in the logistic model having acceptable values since values up to 7000 g have been previously reported for commercial strains of broilers raised to maturity [52]. Similarly, Atil et al. [53] demonstrated that the Logistic model provided better biological estimates than Gompertz and von Bertalanffy models when fitting growth curves for two commercial broiler genetic lines and both sexes. Another study [54] determined that Gompertz and von Bertalanffy models presented higher asymptotic values than the logistic model in Cobb 500 broilers raised to 49 d. Duan-yai et al. [23] explained that data up to 35 d of age could be insufficient to describe chickens’ growth inflection points required by specific models.

In the present study, hatching BW was overestimated in all models. Some authors [21,55] have indicated that the Gompertz model usually overfits this parameter to improve the fit in the following weeks, while another study [24] showed underfit responses. Since hatching and asymptotic BW parameters are usually over- or under-fitted, it has been proposed that these parameters should not be considered [21,55] for practical interpretation, and relevance should be focused on the ability to predict responses in the subsequent weeks [56,57].

As expected, strong negative correlations were observed between *K* and *ti* estimates. Correlation coefficients for males and females, regardless of model, sex, and efficiency group, indicated that the *Wi* was reached earlier as chickens grew faster. Previous reports [4,24] described similar findings in Ross 308 broilers when using Gompertz, logistic, von Bertalanffy, and weighted least squares models. According to Marcato et al. [19], the maximum BW gain in broilers depends on the protein deposition, driven by genetics, and consequently has a daily limit. A recent report from Dukha et al. [58] showed, in a mechanistic model based on Gompertz for body protein mass, that the value of protein deposition drastically alters the potential mature BW at any step of the growth.

Moreover, the maturity rate or precocity parameter at low-rate values may result in late “maturity”. In contrast, higher values of precocity could result in early maturing or reaching the *Wi* [58], as observed in the present study. Thus, it has been proposed [7,19,59] that the longer the chicken takes to reach the *Wi* in protein deposition, which could be directly related to the growth rate, the more efficiently the chicken will produce lean tissue.

In this study, all models, regardless of sex or efficiency group, resulted in R^2^ values greater than 0.99. Several authors have reported similar results when evaluating growth curves under different conditions [4,5,24,57]. Thus, some reports indicate that the model evaluation should be conducted with caution when using R^2^ as a differentiator since this parameter tends to be high and close among models, which could make the identification of a significantly better model complex [5,24,60]. Traditionally, the model that has provided better interpretability and goodness-of-fit in broilers has been the Gompertz model in the traditional form when the dataset contains growth data for more than six weeks of age, which includes the *Wi* [60,61,62,63,64]. However, in this study, the logistic model provided estimates with better biological meaning despite the Gompertz–Laird model obtaining the best goodness-of-fit metrics and the lowest residuals. It indicated that in the evaluation of the growth of broilers up to 35 d, the Gompertz–Laird model shows better predictability capabilities. In contrast, the logistic model produces better interpretable parameter estimates. Therefore, the purpose of the model and the amount of data according to age should be considered when deciding on the best growth functions to use under commercial conditions.

## 5. Conclusions

In conclusion, under the production conditions in which these data were collected, both male and female Ross 308 AP broilers presented a better feed efficiency at 35 d when controlled feeding at 25.45 and 22.50% from the genetic line FI recommendation was established. This FI control program was established between the second and the fourth week of age, with a subsequent increase in the feed allowance during the last week. Additionally, all models demonstrated a reduced growth rate during the second and third week of age, resulting in heavier chickens at 35 d with better feed efficiency and greater BW gain after 24 d for males and 22 d for females. Based on the interpretability of the predicted data and good fit, the logistic model was the best model for commercial growth broiler data up to 35 days of age. Still, the best predictive power was observed with the Gompertz–Laird model.

## Figures and Tables

**Figure 1 animals-13-02447-f001:**
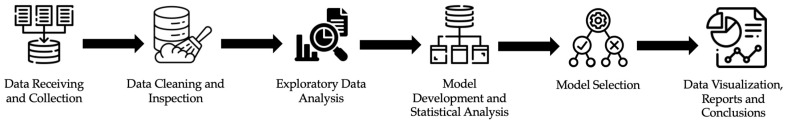
The flow of data analysis.

**Figure 2 animals-13-02447-f002:**
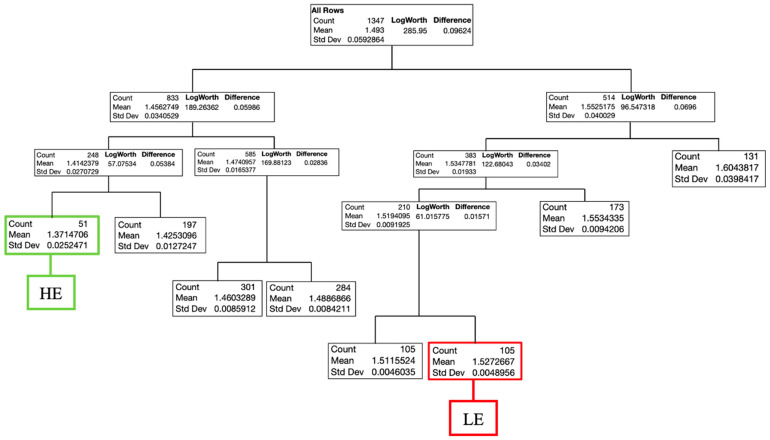
Decision tree for Ross 308 AP male chicken flocks with FCR at 35 d as the response. (R^2^ = 0.93; RASE = 0.016; AICc = −7349.9). LE = low efficiency; HE = high efficiency.

**Figure 3 animals-13-02447-f003:**
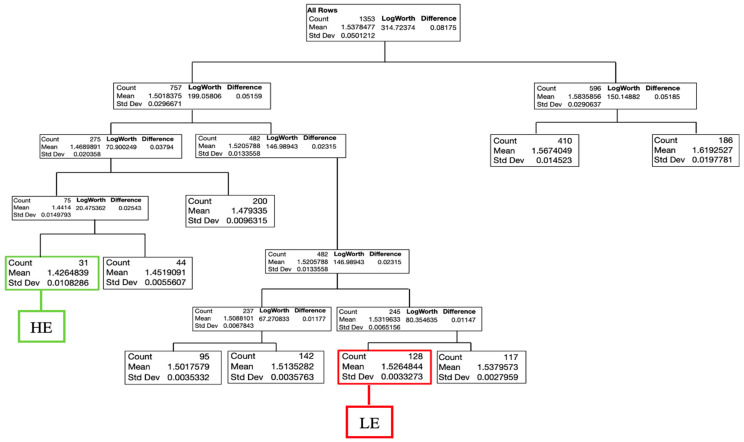
Decision tree for Ross 308 AP female chicken flocks with FCR at 35 d as the response. (R^2^ = 0.95; RASE = 0.012; AICc = −8162.3); LE = low efficiency; HE = high efficiency.

**Figure 4 animals-13-02447-f004:**
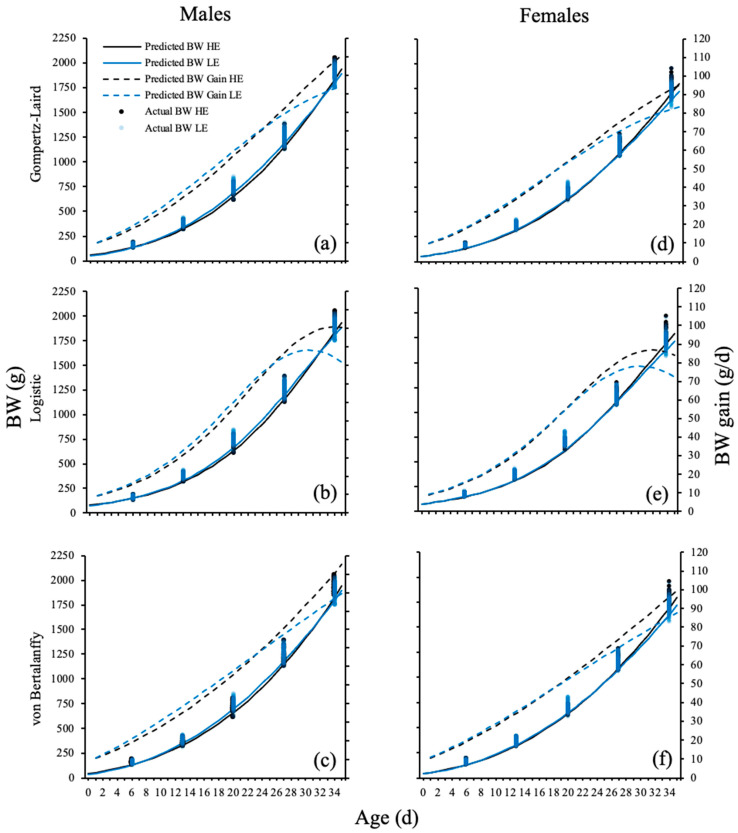
Actual and predicted BW from Gompertz–Laird, logistic, and von Bertalanffy models fit all data by efficiency group ^1^ of Ross 308 AP male (**a**–**c**) and female (**d**–**f**) chickens up to 35 d. LE = low efficiency; HE = high efficiency. ^1^ Feed efficiency groups were determined by decision trees using FCR at 35 d as the response.

**Figure 5 animals-13-02447-f005:**
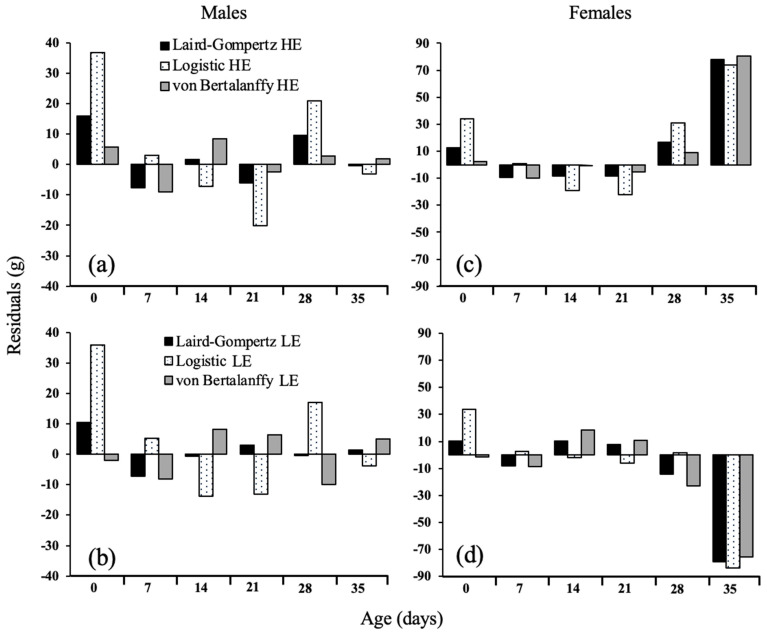
The residuals of Gompertz–Laird, logistic, and von Bertalanffy models fit all data by efficiency group ^1^ of Ross 308 AP male (**a**,**b**) and female (**c**,**d**) chickens up to 35 d. ^1^ Feed efficiency groups were determined by decision trees using FCR at 35 d as the response. LE = low efficiency; HE = high efficiency.

**Table 1 animals-13-02447-t001:** Observed weekly BW, FI, FCR, mortality, and cumulative mortality of male and female low-efficiency (LE) and high-efficiency (HE) groups ^1^ of Ross 308 AP chickens under commercial conditions and feed control.

Age (d)	Males					Females				
	LE	HE	SEM	CV	*p*-Value	LE	HE	SEM	CV	*p*-Value
	(*n* = 51)	(*n* = 105)				(*n* = 31)	(*n* = 128)			
**BW**	--------------(g)------------			(%)		--------------(g)------------			(%)	
0	39 ^b^	41 ^a^	0.3	6.1	<0.001	40	41	0.3	6.5	0.103
7	164	165	1.3	6.5	0.885	161	162	1.6	7.4	0.847
14	381 ^a^	359 ^b^	2.6	5.9	<0.001	362 ^a^	352 ^b^	3.3	6.8	0.038
21	748 ^a^	716 ^b^	4.9	5.5	<0.001	700	692	5.2	5.6	0.310
28	1267 ^a^	1224 ^b^	7.0	4.7	<0.001	1160	1176	7.5	4.8	0.161
35	1891 ^b^	1937 ^a^	7.7	3.4	<0.001	1716 ^b^	1795 ^a^	9.1	3.9	<0.001
**FI**	--------------(g)------------					--------------(g)------------				
7	145 ^a^	138 ^b^	1.4	7.9	0.001	142 ^a^	137 ^b^	1.4	7.6	0.014
14	450 ^a^	406 ^b^	3.1	5.9	<0.001	424 ^a^	396 ^b^	3.8	6.8	<0.001
21	991 ^a^	895 ^b^	5.5	4.8	<0.001	921 ^a^	875 ^b^	5.7	4.7	<0.001
28	1804 ^a^	1616 ^b^	8.8	4.2	<0.001	1646 ^a^	1587 ^b^	9.1	4.1	<0.001
35	2881 ^a^	2655 ^b^	11.7	3.5	<0.001	2620 ^a^	2560 ^b^	13.7	3.9	0.004
**FCR**	------------(g:g)------------					------------(g:g)------------				
7	0.882 ^a^	0.841 ^b^	0.007	6.4	<0.001	0.883 ^a^	0.846 ^b^	0.009	7.3	0.004
14	1.179 ^a^	1.131 ^b^	0.006	4.0	<0.001	1.172 ^a^	1.126 ^b^	0.005	3.4	<0.001
21	1.327 ^a^	1.251 ^b^	0.006	3.9	<0.001	1.317 ^a^	1.265 ^b^	0.005	3.1	<0.001
28	1.424 ^a^	1.322 ^b^	0.005	2.7	<0.001	1.419 ^a^	1.350 ^b^	0.004	2.3	<0.001
35	1.527 ^a^	1.371 ^b^	0.002	1.0	<0.001	1.526 ^a^	1.426 ^b^	0.001	0.4	<0.001
**Week mortality**	-------------(%)------------					-------------(%)------------				
0–7	0.45	0.35	0.03	30.2	0.055	0.49	0.71	0.06	36.3	0.084
8–14	0.81 ^a^	0.59 ^b^	0.05	23.9	0.001	0.75	0.98	0.07	29.2	0.071
15–12	0.59	0.51	0.04	26.6	0.072	0.49	0.54	0.05	32.9	0.846
22–28	0.51	0.46	0.04	30.6	0.269	0.37	0.42	0.05	35.2	0.551
29–35	0.51	0.47	0.03	27.9	0.521	0.38	0.40	0.05	39.6	0.869
**Cum. mortality**	-------------(%)------------					-------------(%)------------				
0–14	1.22 ^a^	0.91 ^b^	0.08	23.1	<0.001	1.20	1.55	0.11	26.1	0.136
0–21	1.77 ^a^	1.39 ^b^	0.11	22.8	0.001	1.78	1.93	0.17	30.1	0.975
0–28	2.34	2.01	0.15	25.2	0.061	2.15	2.72	0.22	30.5	0.223
0–35	3.03	2.93	0.23	28.6	0.415	2.59	3.09	0.27	31.5	0.351

^1^ Feed efficiency groups were determined by decision trees using FCR at 35 d as the response. BW = body weight; FI = feed intake; FCR = feed conversion ratio. ^a,b^ Means in columns followed by different superscript letters are statistically different according to the Student’s *t*-test (*p* < 0.05).

**Table 2 animals-13-02447-t002:** Logistic, Gompertz–Laird, and von Bertalanffy parameter estimates of male and female low-efficiency (LE) and high-efficiency (HE) groups ^1^ of Ross 308 AP chickens under commercial conditions and feed control.

Model Estimate	Males					Females				
	LE (*n* = 51)	HE (*n* = 105)	SEM	CV	*p*-Value	LE (*n* = 31)	HE (*n* = 128)	SEM	CV	*p*-Value
**Logistic**				--(%)--					--(%)--	
Asymptotic response (*W_A_*), g	2934 ^b^	3655 ^a^	41	10.9	<0.001	2657 ^b^	3020 ^a^	38	10.3	<0.001
Exponential growth rate (*K*)	0.122 ^a^	0.114 ^b^	0.001	5.2	<0.001	0.119	0.117	0.001	5.3	0.062
Age at the inflection point (*ti*), d	30 ^b^	34 ^a^	0.2	6.3	<0.001	30 ^b^	32 ^a^	0.3	6.2	<0.001
Response at the inflection point (*Wi*), g	1467 ^b^	1828 ^a^	20	10.9	<0.001	1329 ^b^	1510 ^a^	19	10.3	<0.001
Maximum increment (*MI*), g/d	89 ^b^	103 ^a^	1	6.7	<0.001	79 ^b^	88 ^a^	1	7	<0.001
**Gompertz–Laird**										
Initial response (*W*_0_), g	50 ^b^	57 ^a^	1	14.8	<0.001	51	52	1	14.7	0.221
Initial growth rate (*L*)	0.188 ^a^	0.163 ^b^	0.002	10.9	<0.001	0.182 ^a^	0.172 ^b^	0.003	10.8	0.011
Rate of exponential decay (*K*)	0.038 ^a^	0.029 ^b^	0.001	14.4	<0.001	0.038 ^a^	0.033 ^b^	0.001	13.9	<0.001
Age at the inflection point (*ti*), d	43 ^b^	62 ^a^	1	20.5	<0.001	43 ^b^	51 ^a^	1	17.9	<0.001
Asymptotic response (*W_A_*), g	7717 ^b^	18,730 ^a^	832	62.7	<0.001	6884 ^b^	10,167 ^a^	423	41.9	<0.001
**von Bertalanffy**										
Asymptotic response (*W_A_*), g	143,570 ^b^	678,172 ^a^	80,956	220.5	<0.001	92,429 ^b^	370,448 ^a^	49,854	243.2	<0.001
Integration constant (*B*)	0.911 ^b^	0.940 ^a^	0.004	2.6	<0.001	0.899 ^b^	0.930 ^a^	0.004	3.1	<0.001
Maximum relative growth (*K*)	0.010 ^a^	0.006 ^b^	0.001	42.9	<0.001	0.010 ^a^	0.006 ^b^	0.001	44.7	<0.001
Age at the inflection point (*ti*), d	133 ^b^	251 ^a^	17	68.9	<0.001	122 ^b^	205 ^a^	13	63.9	<0.001
Response at the inflection point (*Wi*), g	42,539 ^b^	200,940 ^a^	23,987	220.5	<0.001	27,386 ^b^	109,762 ^a^	14,771	243.2	<0.001

^1^ Feed efficiency groups were determined by decision trees using FCR at 35 d as the response. ^a,b^ Means in columns followed by different superscript letters are statistically different according to the Student’s *t*-test (*p* < 0.05).

**Table 3 animals-13-02447-t003:** Goodness-of-fit metrics from Logistic, Gompertz–Laird, and von Bertalanffy fit all data of male and female low-efficiency (LE) and high-efficiency (HE) groups ^1^ of Ross 308 AP chickens raised under commercial conditions and feed control.

Model	Males		Females	
	LE (*n* = 51)	HE (*n* = 105)	LE (*n* = 31)	HE (*n* = 128)
** *Logistic* **				
R^2^	0.995	0.996	0.994	0.996
RMSE	44.7	43.0	44.9	41.4
AIC	6834.7	3116.1	7968.6	1920.7
BIC	6852.6	3131.0	7987.2	1933.6
Number of iterations	6	4	5	4
** *Gompertz-Laird* **				
R^2^	0.996	0.997	0.995	0.996
RMSE	41.2	39.2	41.6	38.5
AIC	6729.0	3061.0	7853.1	1893.5
BIC	6746.9	3075.8	7871.7	1906.4
Number of iterations	5	6	5	5
** *von Bertalanffy* **				
R^2^	0.996	0.997	0.995	0.996
RMSE	41.5	38.7	41.7	38.5
AIC	6737.4	3052.5	7854.2	1893.9
BIC	6755.4	3067.3	7872.7	1906.8
Number of iterations	22	197	22	44

^1^ Feed efficiency groups were determined by decision trees using FCR at 35 d as the response.

## Data Availability

The data used in this study is the property of Grupo BIOS Inc., Colombia, and is, therefore, not publicly available.

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
