# Peer review of "Data Analytics of Broiler Growth Dynamics and Feed Conversion Ratio of Broilers Raised to 35 d under Commercial Tropical Conditions"

_animals, 2023, doi:10.3390/ani13152447_

Round 1

Reviewer 1 Report

The main issue addressed in the research is the possibilities to optimize the feed conversion ratio in broilers through mathematical modelling. As it is known the feed conversion is one of the traits that is essential from economic point of view. The authors have constructed their models through exhaustive analysis of data in both male  and female Ross 308 broilers that are raised in tropical climate.  The topic is relevant and original and such studies deserve attention since they help to improve the decision making process for managers in poultry, particularly broiler industry. My main remarks however concern section Material and methods. Can the authors provide more information how the body weight and feed intake have been measured? It remains unclear for the reader. Were the measurement taken daily or weekly? The conclusions are derived from the results and are concisely presented. The number of tables and figures is adequate. 

The quality of the English language is good.

Author Response

Thank you for your comment. The sample size and frequency were clarified in lines 101 – 104.

“Weekly BW corresponds to averages of samples of 1% of the population per broiler house, as it is usually evaluated in industrial settings. Feed was weighed and offered according to the feed allocation plan designed for each farm.”

Reviewer 2 Report

The work entitled "Data analytics of broiler growth dynamics and feed conversion ratio of broilers raised to 35 d under commercial tropical conditions" is an interesting work presenting the application of data analysis to real-life problems.

Firstly, the Authors give the study's background, justifying the research's need and explaining their attitude. In the Materials and Methods section, the Authors present the database and provide the methods used in the experiment. Even though the procedure was well described, the section lacks a scheme, which would make it easier for the reader to orientate himself in the course of the experiment quickly. The Authors also did not indicate from which p-value the differences are considered statistically significant and did not explain why they used such criteria as the coefficient of determination, Akaike's information criterion (AIC), Bayesian information criterion (BIC), number of iterations, and root mean square error (RMSE), to assess achieved models. One sentence justifying the choice of models used would also be helpful.

The results are well documented. Correct statistical models were used for the analysis. However, the authors could present some of the data (e.g., from Table 1) in the form of graphs, allowing for a better presentation of the results. It would also be recommended to explain the abbreviations used in the tables and figures in their descriptions (or as footnotes).

The discussion is conducted correctly. The Authors mainly refer to literature from recent years.

The quality of the language is correct, only a slight adjustment is required.

Author Response

C: The section lacks a scheme, which would make it easier for the reader to orientate himself in the course of the experiment quickly.

A: Thanks for the comment. It is important to clarify that this paper is not an experiment or collection of experiments but an actual evaluation of records of broiler production conditions. Subsequently, by definition, commercial data lacks a “scheme” or data structure. In the world of data analytics, this type of databases are called unstructured data. However, to clarify more the procedures the Figure 1 was added (Line 122-123).

C: The Authors also did not indicate from which p-value the differences are considered statistically significant

A: Thanks for noticing, it was indicated in lines 174-175

C: and did not explain why they used such criteria as the coefficient of determination, Akaike's information criterion (AIC), Bayesian information criterion (BIC), number of iterations, and root mean square error (RMSE), to assess achieved models. One sentence justifying the choice of models used would also be helpful.

A: Parameters were selected following the same procedure than other authors to make this study comparable with previous reports. Some references were added in line 173.

C: The results are well documented. Correct statistical models were used for the analysis. However, the authors could present some of the data (e.g., from Table 1) in the form of graphs, allowing for a better presentation of the results. It would also be recommended to explain the abbreviations used in the tables and figures in their descriptions (or as footnotes).

A: Thanks for the comment. Abbreviations were added in the title of tables or graphs or as footnotes. According to the previous recommendation, a scheme was added to the paper which increased the number of figures. Then, keeping Table 1 as it currently is presented would help to maintain all the data summarized.

Reviewer 3 Report

In this manuscript, the authors compared growth patterns of male and female Ross 308 AP broiler chickens raised under commercial tropical conditions with controlled feeding to determine conditions that optimize feed conversion ratio. The Articles is well written. It scientifically addresses valuable topics, and extends the current literature by employing a multivariate approach. Logistics, Gompertz-Laird and von Bertalanffy's data on week BW represent an important commitment, and I think this is the most powerful and valuable aspect of the article.

However, the authors are left with a few issues that should be seriously considered. I suggest revisions, with an emphasis on attention to details such as these, to maximize the potential impact of the work.

Specific comments are as follows.

Line 29 "Placing" is so unofficial in the manuscript...hope the author pays more attention to improving the overall English grammar and colloquial accuracy, not only the revision I mentioned here.

Line 33 abbreviation: the abbreviation between the full word brackets is used at the first time, after that only the abbreviation should be used.

Lines 61-62 This is interesting. Please be more specific. What explains these contradictory results? How will these conflicting results affect the conclusions and limitations of other studies.

Lines 69-71 add a reference to this statement.

Line 93 still need to provide a detailed text and description of experiments in the database.

Line 112-116 moved to "2.2 Statistical Analysis"

Lines 172-173 What method/reference is needed for Correlation and Regression Analysis'?

Lines 179-180 "clasified the flocks into the HE group" This is an awkward sentence, hard to understand. should be rewritten.

Line 247 What does "lower (P < 0.001) ti and WA" mean? Also, I suggest streamlining it and removing any unnecessary details.

Lines 329-330 Although there is nothing to mention, some information, as written in this sentence, is irrelevant to the rest of the manuscript. I suggest deleting or clarifying the sentence.

Lines 337-338 What does this mean?

References: Please revise the References section to comply with journal guidelines.

Lastly, I recommend a review of the written language (correcting grammatical errors and formatting errors, e.g. as word spacing). Authors need to carefully revise the text to reach a wide audience. Overall, I found this to be an important and interesting study. Well deserved to be published in Animals.

Author Response

C: Line 29 "Placing" is so unofficial in the manuscript...hope the author pays more attention to improving the overall English grammar and colloquial accuracy, not only the revision I mentioned here.

A: Thanks for the comment. Although the word “Placed” comes from the placement of chickens, used in poultry it was replaced for “housed”.

C: Line 33 abbreviation: the abbreviation between the full word brackets is used at the first time, after that only the abbreviation should be used.

A: Abbreviations were corrected and defined as indicated by the author’s guidelines. In the abstract, the main text and figures and tables

C: Lines 61-62 This is interesting. Please be more specific. What explains these contradictory results? How will these conflicting results affect the conclusions and limitations of other studies.

Thanks for noticing this statement that, in fact, it is explained detailed with references in the subsequent lines. This explanation was done to clarify the differences in feeding systems in the world (ad libitum and restricted). These statements explained the contrast between ad libitum and controlled feeding that was used in this particular broiler operation.

C: Lines 69-71 add a reference to this statement.

 A: Some references were added to the paragraph highlighting previous reports under experimental conditions.

C: Line 93 still need to provide a detailed text and description of experiments in the database.

A: Although this is not an experiment, these are actual field conditions, all description of the broiler operation is detailed in materials and methods. Number and type of records (lines 95-97), number of chickens included in the report, number of farms and timeframe of data collected (lines 104-107), management conditions (107-112). This study corresponds to a data analytics work, not a metanalysis of experiments.

C: Line 112-116 moved to "2.2 Statistical Analysis"

A: These lines described how data was organized and a procedure before the statistical analysis. Thank you for the comment. Sections of the lines were reorganized in lines 118-120.

C: Lines 172-173 What method/reference is needed for Correlation and Regression Analysis'?

A: References of papers that use similar analyses were listed. These correlation and regression analysis are already standard methods use by several authors. No new methods were developed for this paper. 

C: Lines 179-180 "clasified the flocks into the HE group" This is an awkward sentence, hard to understand. should be rewritten.

A: Sentence clarified in lines 198-202

C: Line 247 What does "lower (P < 0.001) ti and WA" mean? Also, I suggest streamlining it and removing any unnecessary details.

A: An explanation to this result was added in lines 269-271.

C: Lines 329-330 Although there is nothing to mention, some information, as written in this sentence, is irrelevant to the rest of the manuscript. I suggest deleting or clarifying the sentence.

A: This information is necessary to clarify the feed restriction since different strategies have been described in the literature. Some feed restrictions can last a couple of days or during specific weeks of age while our study depicts a continuous feed restriction during the entire life cycle.  

C: Lines 337-338 What does this mean?

 It was clarified in lines 363-364.

C: References: Please revise the References section to comply with journal guidelines.

All references were carefully double-checked.